# Inheritance of OCT4 predetermines fate choice in human embryonic stem cells

Samuel C Wolff[1], Katarzyna M Kedziora[1], Raluca Dumitru[1], Cierra D Dungee[1], Tarek M Zikry[2], Adriana S Beltran[1], Rachel A Haggerty[3], JrGang Cheng[4], Margaret A Redick[1] & Jeremy E Purvis[1,3,5,*]

## Abstract

It is well known that clonal cells can make different fate decisions, but it is unclear whether these decisions are determined during, or before, a cell's own lifetime. Here, we engineered an endogenous fluorescent reporter for the pluripotency factor OCT4 to study the timing of differentiation decisions in human embryonic stem cells. By tracking single-cell OCT4 levels over multiple cell cycle generations, we found that the decision to differentiate is largely determined before the differentiation stimulus is presented and can be predicted by a cell's preexisting OCT4 signaling patterns. We further quantified how maternal OCT4 levels were transmitted to, and distributed between, daughter cells. As mother cells underwent division, newly established OCT4 levels in daughter cells rapidly became more predictive of final OCT4 expression status. These results imply that the choice between developmental cell fates can be largely predetermined at the time of cell birth through inheritance of a pluripotency factor.

**Keywords** cell fate; human embryonic stem cells; OCT4; pluripotency; single-cell dynamics

**Subject Categories** Development & Differentiation; Quantitative Biology & Dynamical Systems; Stem Cells

**Mol Syst Biol. (2018) 14: e8140**

## Introduction

It is well established that clonal cells can make distinctly different fate decisions (Suda *et al*, 1984a,b; Suel *et al*, 2006). An important conceptual challenge, however, is to understand to what extent a cell exerts independent control over its own fate (Symmons & Raj, 2016). At one extreme, a cell's fate may be entirely shaped through environmental stimuli and its own intracellular signaling cues; at the other extreme, a cell's fate is already determined before it emerges from its mother cell, rendering it impervious to external cues.

Multiple studies have linked different cell fate decisions to differences in single-cell signaling patterns (Purvis *et al*, 2012; Albeck *et al*, 2013; Lin *et al*, 2015; Lane *et al*, 2017). However, tracking cells over multiple cell cycle generations suggests that such intracellular signals can themselves be inherited from mother to daughter cells (Sigal *et al*, 2006). For example, the response to death ligands in human cells is similar among sister cells through inheritance of apoptotic protein factors (Spencer *et al*, 2009). In addition, cell cycle checkpoint decisions in daughter cells were shown to be influenced by the signaling history of the mother cell (Arora *et al*, 2017; Barr *et al*, 2017; Yang *et al*, 2017). By considering the histories of individual cells, these studies have eroded the concept that cells can make fully autonomous fate decisions and raise the question of what mechanisms may regulate the inheritance of fate-determining factors.

How fate choice is controlled by inherited factors is an important question in stem cell biology. During human development, proliferating stem cells give rise to complex and heterogeneous tissues through a dynamic interplay of intracellular signaling events, cell–cell communication, and morphogen gradients (Deglincerti *et al*, 2016; Etoc *et al*, 2016). *In vitro* imaging of attached human embryos has yielded unprecedented insights into the single-cell patterning of the human gastrula (Deglincerti *et al*, 2016). However, because these cells must be necessarily fixed in preparation for imaging, it is not possible to follow any given cell over the course of its cellular lifetime. It is therefore difficult to pinpoint precisely when an individual cell makes the decision to differentiate, how fate-determining factors are inherited from mother to daughter cell, or why two closely related cells may choose different fates. As an alternative approach, human embryonic stem cells (hESCs) represent a promising system for studying embryonic cell fate decisions in real time (Thomson *et al*, 1998; Bernardo *et al*, 2011; Nemashkalo *et al*, 2017). hESCs can be maintained indefinitely in cell culture and are amenable to introduction of fluorescent biosensors to report on intracellular signaling activity (Nemashkalo *et al*, 2017). These features make it possible for hESCs to be used to understand how fate choice is determined among proliferating stem cells.

1 Department of Genetics, University of North Carolina, Chapel Hill, Chapel Hill, NC, USA
2 Department of Biostatistics, University of North Carolina, Chapel Hill, Chapel Hill, NC, USA
3 Curriculum for Bioinformatics and Computational Biology, University of North Carolina, Chapel Hill, Chapel Hill, NC, USA
4 UNC Neuroscience Center, University of North Carolina, Chapel Hill, Chapel Hill, NC, USA
5 Lineberger Comprehensive Cancer Center, University of North Carolina, Chapel Hill, Chapel Hill, NC, USA
*Corresponding author. Tel: +1 919 962 4923; E-mail: jeremy_purvis@med.unc.edu

In this study, we developed an endogenous fluorescent reporter for the human pluripotency factor OCT4 to study its inheritance over multiple cell cycle generations. We conducted time-lapse fluorescence imaging of hESCs during differentiation to extraembryonic mesoderm and followed the signaling behaviors of individual cells until final fate decisions were determined. We found that the decision to differentiate is largely determined before the differentiation stimulus is presented to cells and can be predicted by a cell's preexisting OCT4 signaling patterns. Further, we found that OCT4 levels were heritable from mother to daughter cell and that OCT4 levels established in newly born daughter cells were strongly predictive of long-term cellular states. These results suggest that the choice between two developmental fates can be strongly predetermined within a short time after cell birth.

## Results

We first established an experimental system for generating both self-renewing and differentiating cells in response to the same developmental signal (Fig 1A). When treated with bone morphogenetic protein 4 (BMP4) for 24 h, a subpopulation of hESCs showed reduced expression of the core pluripotency factor OCT4 and accumulation of the caudal type homeobox 2 (CDX2) transcription factor (Fig 1B). Quantitative immunofluorescence (IF) revealed two distinct populations of cells: a pluripotent population with low CDX2 expression that retained the ability to differentiate into other cell types (Appendix Fig S1); and a differentiating population of cells with reduced OCT4 expression, increased CDX2 expression, and enlarged morphology (Fig 1C). Although BMP4 treatment was originally reported to initiate differentiation toward the trophoblast lineage (Xu *et al*, 2002), further study has revealed that, in the presence of fibroblast growth factor, it induces markers such as *BRACHYURY* and *ISL1* that are more closely associated with extraembryonic mesoderm (Bernardo *et al*, 2011). We confirmed the expression of mesodermal markers in BMP4-treated hESCs through quantitative PCR (Appendix Fig S2). Thus, treatment of hESCs with BMP4 triggered a binary cell fate decision within 24 h.

To understand how and when individual hESCs make this decision, we developed a fluorescent reporter system to monitor expression of the endogenous OCT4 protein, a canonical marker of the pluripotent state (Nichols *et al*, 1998). We used CRISPR-mediated genome editing to fuse a monomeric red fluorescent protein (mCherry) to the endogenous OCT4 allele in WA09 (H9) hESCs and

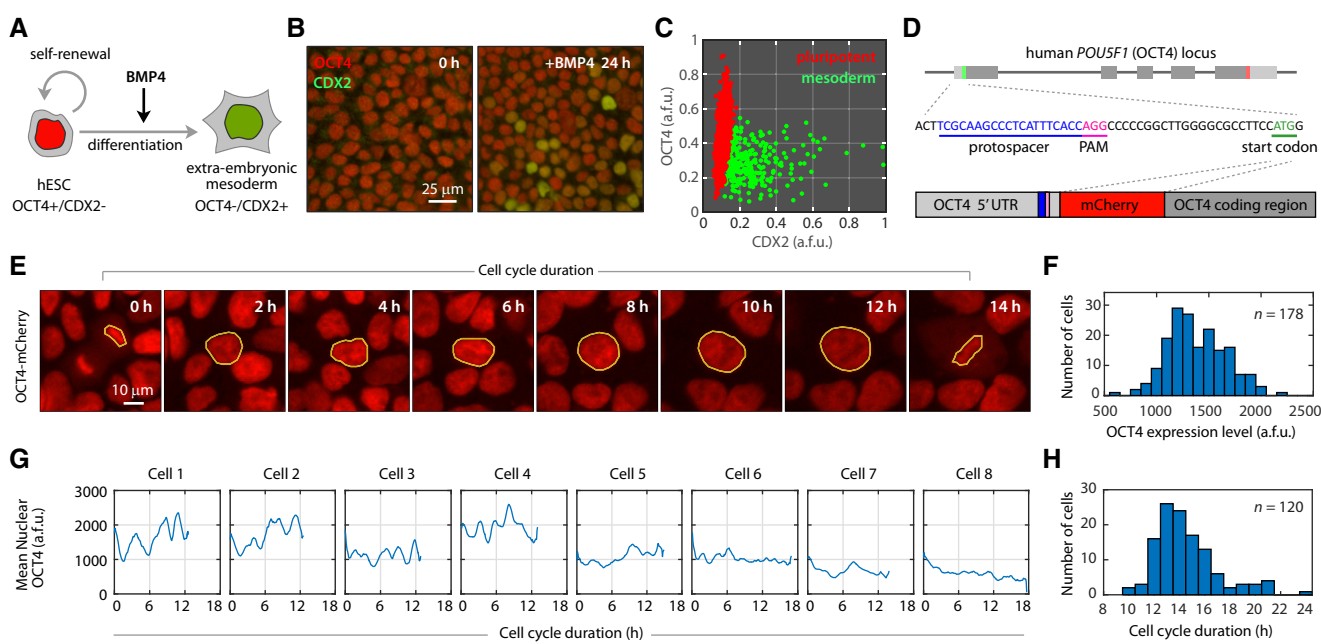

**Figure 1. Single-cell dynamics of OCT4 in human embryonic stem cells.**

A    Individual hESCs have the potential to generate another stem cell through self-renewal or to differentiate into a more lineage-specific cell type.

B    Before differentiation, hESCs show uniformly high expression of OCT4. Treatment with BMP4 produces a mixture of OCT4+/CDX2− self-renewing hESCs and OCT4−/CDX2+ mesodermal cells.

C    Quantification of OCT4 and CDX2 expression by immunofluorescence after 24 h of BMP4 treatment reveals two populations of hESCs. Cells were assigned to one of two distinct populations based on a two-component mixed Gaussian distribution (see Appendix Fig S6).

D    A fluorescent mCherry coding sequence was introduced into the endogenous OCT4 locus of H9 hESCs using CRISPR-mediated homologous recombination.

E    Filmstrip of OCT4 dynamics in an undifferentiated hESC throughout its cell cycle duration. Yellow outlines indicate the region used to quantify mean nuclear fluorescence intensity.

F    Distribution of OCT4 levels in individual hESCs. A single OCT4 level was quantified for each cell by averaging the mean nuclear mCherry intensity over the lifetime of the cell.

G    Single-cell traces of OCT4 signaling. The length of each cell's trace indicates its cell cycle duration.

H    Distribution of cell cycle durations for 120 hESCs.

isolated a clonal population of single-allele knock-in reporter cells (Fig 1D and Materials and Methods). The OCT4-mCherry fusion protein showed correct genomic targeting; accurate co-localization with the endogenous OCT4 protein; similar degradation kinetics; and the same chromatin binding pattern near the promoters of OCT4 target genes (Appendix Fig S3). Moreover, cells bearing the OCT4-mCherry reporter were competent to differentiate into multiple differentiated cell types (Appendix Fig S4), and time-lapse imaging did not alter their proliferation characteristics (Appendix Fig S5). For each cell, we calculated a single OCT4 expression level by averaging OCT4-mCherry intensity over its cell cycle duration (Fig 1E and F). In addition, we examined the time-series profile of OCT4 dynamics for individual cells and found that the majority of hESCs (68%) displayed sporadic pulses of OCT4 expression that lasted ~ 1.5 h, with some cells showing as many as seven pulses (Fig 1G). Finally, we calculated individual cell cycle durations, which ranged from 10 to 24 h with a mean duration of 14.6 h

(Fig 1H), consistent with the reported population doubling time of ~ 16 h (Ghule *et al*, 2011). Thus, our reporter system enabled the reliable analysis of single-cell OCT4 dynamics in hESCs and revealed considerable heterogeneity in untreated stem cells.

With this system in place, we set out to capture the fate decisions of hESCs in real time. First, we performed time-lapse fluorescence imaging of H9 OCT4-mCherry hESCs for 42 h under basal conditions to capture multiple complete cell cycles before differentiation (Fig 2A). We then treated these cells with 100 ng/ml BMP4 to induce differentiation while continuing to monitor their responses. Within 12 h of treatment, each cell began to follow one of two distinct fate paths: sustained accumulation of OCT4 or a precipitous decrease in OCT4. After 24 h, cells were fixed and stained for expression of CDX2 to determine their final differentiation status (Fig 2B). We imposed a strict cutoff to classify each cell as either self-renewing or differentiated based on its OCT4 and CDX2 expression levels. By fitting the data in Fig 2B to a two-component

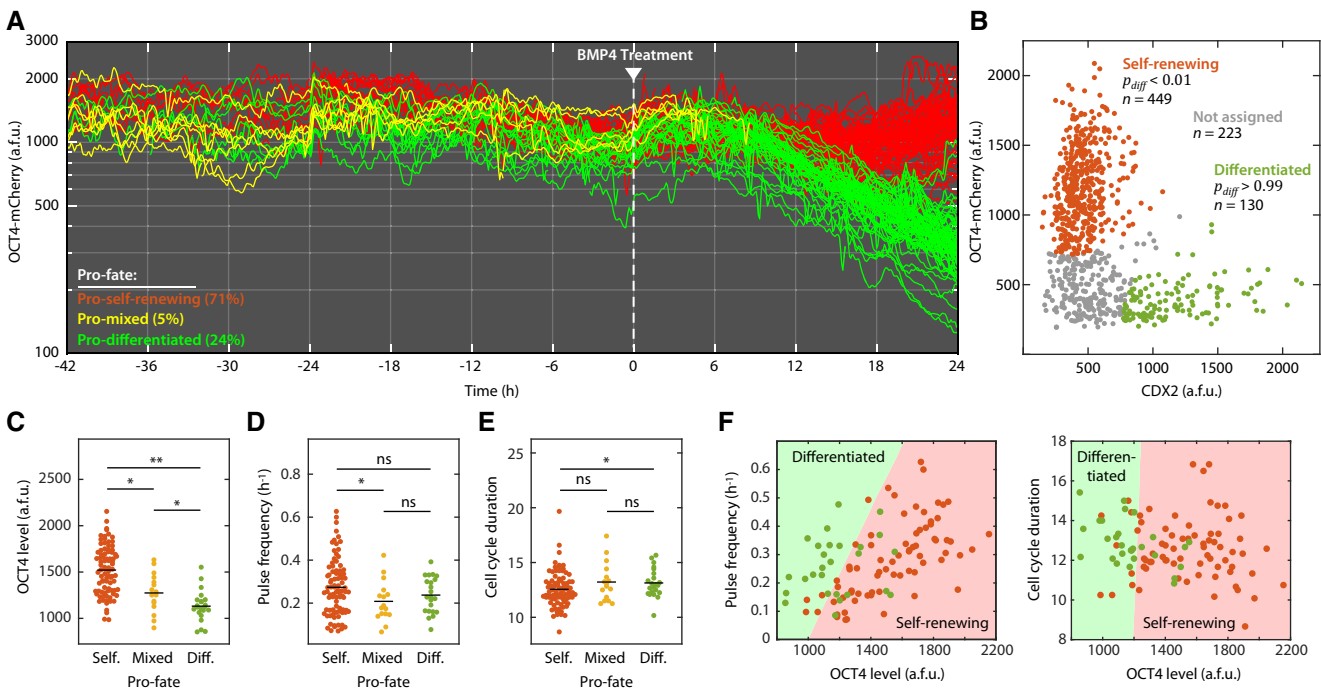

**Figure 2.  Differences in OCT4 dynamics before differentiation stimulus predict eventual fate decisions.**

A   Single-cell traces of hESCs before and after treatment with 100 ng/ml BMP4. Cells were imaged for 42 h prior to BMP4 treatment. Mean nuclear OCT4 levels were quantified every 5 min, and individual cells were tracked from the cell division event that created the cell until its own division.

B   24 h after BMP4 treatment, cells were fixed, stained for expression of CDX2, and returned to the microscope for registration with the final time-lapse image. Mean nuclear OCT4-mCherry and CDX2 were quantified for each cell, and the resulting distribution was fit to a two-component mixed Gaussian distribution representing self-renewing (OCT4$^+$/CDX2$^-$) and differentiated (OCT4$^-$/CDX2$^+$) cells (Appendix Fig S6). hESCs that could be assigned to either distribution with > 99% confidence (red and green dots, $P_{diff} < 0.01$ or $P_{diff} > 0.99$) were considered for pro-fate analysis. Cells that did not reach this threshold (gray dots) were not used to determine pro-fate. Because cells in the final frame were assigned to only self-renewing or differentiated categories, pro-mixed cells (yellow traces in panel A do not persist to the final frame).

C   Distributions of OCT4 levels in pro-self-renewing, pro-mixed, and pro-differentiated cell populations.

D   Distributions of OCT4 pulse frequencies in pro-pluripotent, pro-mixed, and pro-differentiated cell populations.

E   Distributions of cell cycle durations in pro-pluripotent, pro-mixed, and pro-differentiated cell populations. To gain an unbiased look at preexisting determinants of cell fate in panels (D, E), only cells who completed their entire cell cycle duration before BMP4 addition (*t* = 0) were included in the analysis.

F   Decision surfaces for logistic regression classifier. Cells with self-renewing (red) or differentiated (green) pro-fates are plotted above the predicted class surfaces. (left) Slice of decision surface for pulse frequency vs. OCT4 levels at a cell cycle duration of 14 h. (right) Slice of decision surface for cell cycle duration vs. OCT4 level at a pulse frequency of 0.25 h$^{-1}$.

Data information: (C–E) *$P < 0.05$, **$P < 0.0005$, two-sample Kolmogorov–Smirnov test; ns, not significant.

Gaussian distribution (Appendix Fig S6), we selected only those cells that belonged exclusively to either the self-renewing distribution ($P_{self}$ < 0.01) or the differentiated distribution ($P_{diff}$ > 0.99), where $P_{diff}$ represents the probability that a given cell has differentiated.

We then traced both populations back through time—spanning multiple cell division events—and labeled each earlier cell according to its "pro-fate"—the fate to which it (or its progeny) would ultimately give rise. The majority of cells in the tracked population were either pro-self-renewing (71%, red traces in Fig 2A), giving rise to only self-renewing cells, or pro-differentiated (24%, green traces in Fig 2A), giving rise to only differentiated cells. Although the majority of progenitor cells gave rise exclusively to a single fate, approximately 5% of cells were "pro-mixed" and gave rise to both self-renewing and differentiated fates (yellow traces in Fig 2A). This observation suggests that fate decisions were strongly heritable and was further supported by the observation that 89% of sister cells chose the same fate. Thus, time-lapse imaging allowed us to group hESCs by their eventual fate categories before they had received a differentiation signal or had made a clear fate decision.

We next asked whether there were preexisting differences between pro-self-renewing, pro-mixed, and pro-differentiated cell populations that might influence their fate decisions. Indeed, pro-self-renewing cells showed significantly higher OCT4 expression levels than either pro-differentiated or pro-mixed populations (Fig 2C). This result echoes the observation that repression of Oct4 in mouse ESCs induces loss of pluripotency and differentiation to trophectoderm (Niwa *et al*, 2000). Pro-self-renewing cells also showed a modest increase in pulse frequency (number of OCT4 pulses per hour) over pro-mixed cells (Fig 2D). Pro-self-renewing cells also had shorter cell cycle durations, on average, than both pro-mixed and pro-differentiated populations (Fig 2E). The latter finding is consistent with reports that hESC self-renewal is linked with a shortened G1 cell cycle phase (Becker *et al*, 2006; Matson *et al*, 2017).

We then asked whether these single-cell features—mean OCT4 levels, pulse frequency, and cell cycle duration—could predict cell fate. We identified cells that completed a full cell cycle before BMP4 stimulation, extracted their single-cell quantities, and used these predictors to train a model to correctly classify cells according to pro-fate. We tested a variety of classifier models (Materials and Methods and Appendix Fig S7) and found that all models were accurate within a range of 72–86%, which represents the percentage of correctly classified cells. Figure 2F shows the results of a logistic regression model (82% accuracy) employing fivefold cross validation. In all classifier models tested, OCT4 level was the strongest predictor of pro-fate ($P = 3.1 \times 10^{-6}$) followed by burst frequency ($P = 0.0081$); cell cycle duration was not a significant predictor of cell fate ($P = 0.85$; Appendix Table S1). Taken together, these results show that undifferentiated hESCs display heterogeneous OCT4 levels, pulse dynamics, and cell cycle durations. Of these single-cell features, OCT4 level—which was evident more than 1 day and as many as two cell cycles before the differentiation cue was given—was the strongest predictor of stem cell fate.

Because OCT4 level was the strongest predictor of cell fate (Fig 2F), we next asked how heterogeneity in OCT4 levels arises in a population of hESCs. To identify the source of cell-to-cell heterogeneity, we monitored OCT4 expression continuously in proliferating, undifferentiated hESCs for 72 h and generated lineage trees of single-cell relationships (Fig 3A). Visual inspection of the lineages revealed that OCT4 levels were most similar among closely related cells (i.e., cells emerging from a common cell division event), providing further support that OCT4 levels are heritable from mother to daughter cell. To quantify this heritability pattern, we calculated the differences in OCT4 levels between pairs of cells as a function of their shared history. Sister cells showed the most similarity in OCT4 levels, followed by "cousin" and "second cousin" cells (Fig 3B). Both sister and cousin cells, but not second cousins, were more similar than randomly paired cells, indicating that similarity in OCT4 levels can persist for at least two cell cycle generations (Spencer *et al*, 2009). Suspecting that each cell division event introduced variability in OCT4 levels, we detected a significant correlation between the number of cell divisions and the difference in OCT4 levels between all pairs of cells (Appendix Fig S8). Thus, OCT4 levels are heritable from mother to daughter cell, but each division event introduces incremental variability in OCT4 expression levels.

To understand how variability in OCT4 levels arises during cell division, we examined individual division events at high temporal resolution. As cells entered mitosis, OCT4 became visibly associated with the condensed chromosomes (Fig 3C, *left panel*). This compacted state persisted throughout anaphase until the two daughter chromatids could be visibly distinguished (Fig 3C, *center panel*). We used this first time point—before cytokinesis was complete—to quantify the levels of OCT4 in both newly born daughter cells (Fig 3C, *center panel*). Comparison of OCT4-mCherry intensity between daughter cells revealed that OCT4 was not equally allocated during cell division. Instead, the ratio of OCT4 between daughters showed a bell-shaped distribution with a central tendency toward a ratio of 1 ($r = 1:1$; Fig 3D). Approximately 38% of divisions produced daughter cells with $r = 5:6$ or a more extreme ratio; 12% of divisions resulted in $r = 3:4$ or a more extreme ratio; and 3% of division events resulted in $r = 1:2$ or a more extreme ratio.

Multiple observations suggested that cell-to-cell differences in OCT4 measured shortly after cell division were significant. First, the measured differences in OCT4 intensities between sisters were significantly greater than differences in DNA content (Fig 3E). In addition, the ratio of OCT4 established within 15 min of division was significantly correlated with the final OCT4 ratio between sisters (Appendix Fig S9). Further, because the half-life of OCT4 was calculated to be 7.34 h (Appendix Fig S3), it is unlikely that differences in OCT4 in newly born cells were due to stochastic changes in protein production or OCT4 would be > 99% maternal 5 min after division and > 95% maternal after 30 min (see Materials and Methods). Moreover, OCT4 ratios were not correlated with nuclear area or radial position within the colony (Appendix Fig S10). Together, these results show that significant variability in OCT4 protein levels are established within 1 h of cell division.

Thus, variability in OCT4 expression levels arises during cell division through transmission of OCT4 from mother to daughter cells (Fig 3A and B, and Appendix Fig S8) as well as asymmetric distribution of OCT4 to daughter cells (Fig 3C–E). These results suggest that inheritance of OCT4 may play a role in predicting stem cell fate. To gain a better picture of this process, we divided each cell's history into three time periods: "maternal", "inherited", or "autonomous" (Fig 4A). Maternal OCT4 encompasses the OCT4

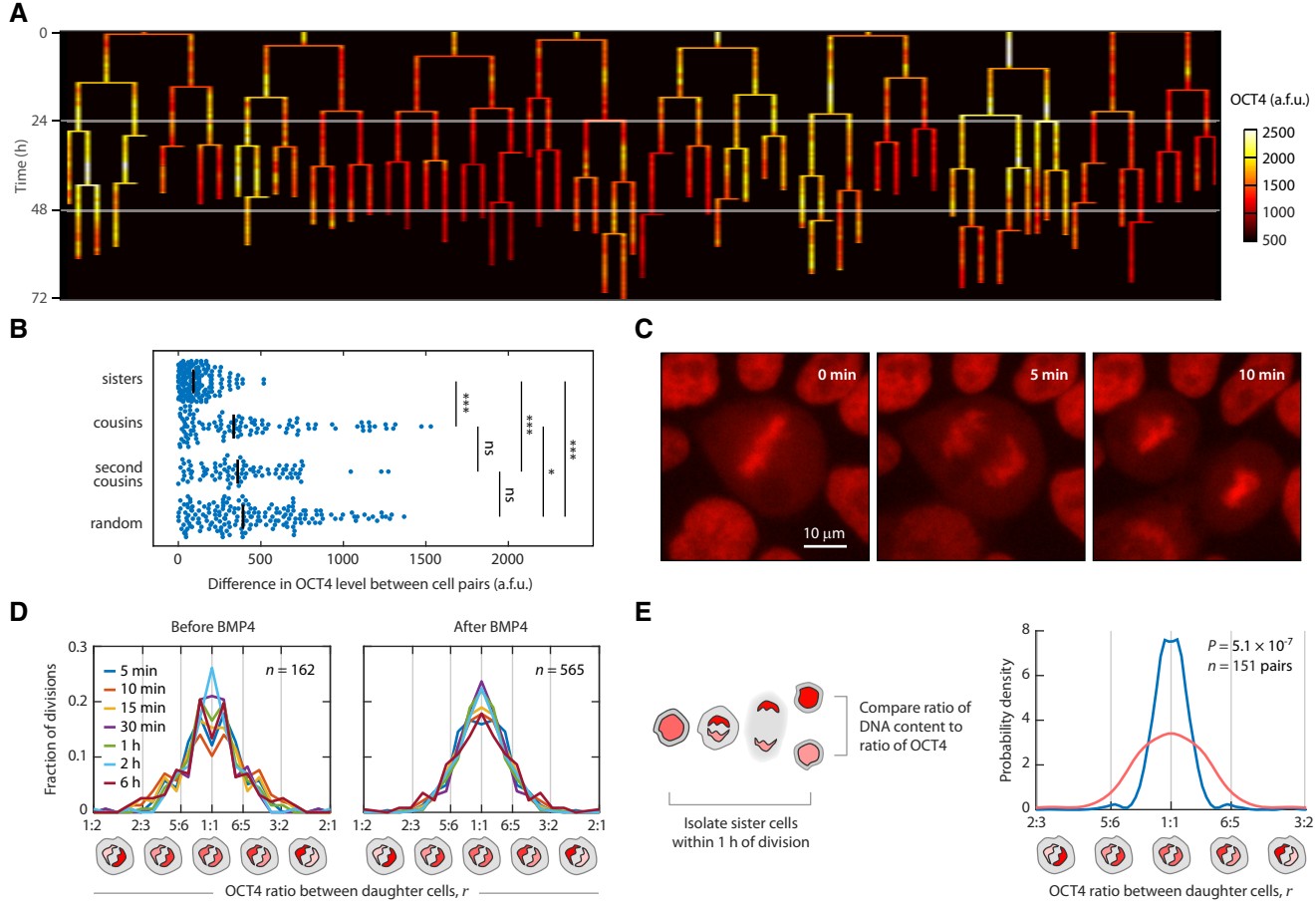

**Figure 3.    Transmission and distribution of maternal OCT4 to daughter cells during cell division.**

A    Lineage of OCT4 expression dynamics. Mean nuclear OCT4 levels were quantified in individual hESCs continuously for 72 h under undifferentiated conditions. Vertical bars represent individual cells that completed an entire cell cycle lifetime. Thin horizontal bars denote cell division events. Color scale indicates low (black), intermediate (red), and high (white) OCT4 expression levels.

B    Differences in OCT4 levels between sister cells, cousin cells, second cousin cells, and randomly paired cells. *$P < 0.05$, ***$P < 0.0005$, two-sample Kolmogorov–Smirnov test; ns, not significant.

C    Filmstrip showing distribution of OCT4 to daughter cells during cell division.

D    Distribution of OCT4 ratios between sister cells before and after BMP4 treatment. Ratios for both sister cells (*r* and *1/r*) are plotted to emphasize central tendency toward *r* = 1:1. Differently colored curves represent the distribution of ratios at different time points after division. For time points after 5 min, the ratio was determined by first calculating a mean OCT4 level for each sister cell among all previous time points and then calculating the resulting ratio between sisters.

E    Proliferating H9-OCT4-mCherry hESCs were imaged for 24 h, fixed, and stained for DNA content with DAPI. The probability density of DAPI ratio (blue) and OCT4-mCherry ratio (red) between sister cells that divided ≤ 1 h prior to fixation was calculated for 151 cell pairs using a normal kernel function. A two-sample Kolmogorov–Smirnov test was used to determine significance.

expression dynamics in the mother cell up to the moment of cell division. Inherited OCT4 levels are those established within 1 h after cell division. This time period reflects both the amount of OCT4 transmitted to both daughters as well as any asymmetric distribution between daughters. Lastly, we defined autonomous OCT4 as changes in OCT4 expression that occurred during the remainder of the daughter cell's own lifetime.

We reasoned that if significant variability in OCT4 is introduced during division, we would expect to see "mixing" of OCT4 expression levels just before cell division (Sigal *et al*, 2006). Figure 4B shows how 30 randomly chosen cells—color-ranked according to final OCT4 expression—experience mixing as a function of the cells' histories. As predicted, significantly more mixing occurred during the maternal period, just before cell division, whereas

relatively little mixing occurred after cell division. To quantify this phenomenon more fully, we aligned mother–daughter cell pairs to the time of division and, at each aligned time point, calculated the coefficient of determination ($R^2$) between the current OCT4 level and the "final" OCT4 level (8 h after division; Fig 4C, Dataset EV1 and accompanying code). The time point of 8 h was chosen as both a long-term indication of a cell's OCT4 status and a lower bound on the distribution of observed cell cycle lifetimes (Fig 2E). This analysis provides a continuous quantification of how much the variance in final OCT4 levels can be explained as a function of the cellular history. We found that differences in total OCT4 expression among mother cells explained ~ 15% of variance in final OCT4 levels. This percentage was relatively steady throughout the maternal history up to the time of cell division. During the

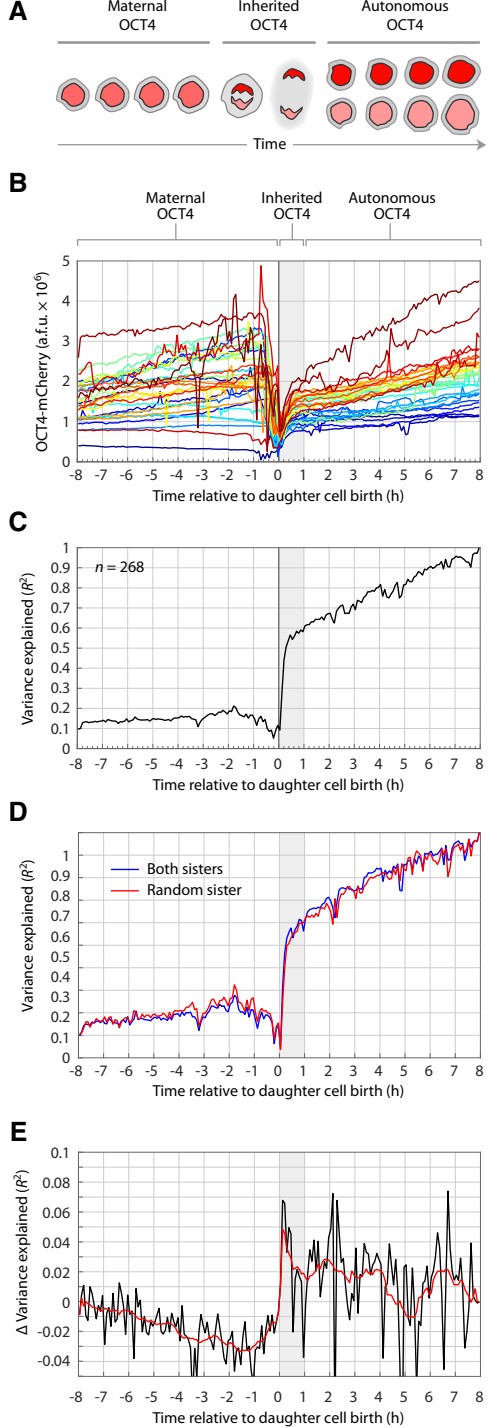

**Figure 4. OCT4 levels established during cell division predict long-term OCT4 levels.**

A Single-cell histories of OCT4 dynamics were classified into three time periods. Maternal OCT4 encompasses the levels throughout the lifetime of the mother cell. Inherited OCT4 represents the levels established within 1 h of division. Autonomous OCT4 refers to the OCT4 dynamics during the remaining lifetime of the daughter cell.

B OCT4 mixing as a function of cell history. Thirty randomly chosen cells were colored according to their OCT4 expression level at 8 h after division.

C Variance in final OCT4 expression explained over time during maternal, inherited, and autonomous periods of cell history. At each time point, the coefficient of determination ($R^2$) between total OCT4 expression levels and final OCT4 expression at 8 h was calculated.

D Estimate of variance explained by asymmetric distribution of OCT4 to daughter cells. The same $R^2$ over time as in panel (C), except that mother–daughter traces were randomly sampled from a set of mother–daughter traces that contained sister-cell pairs (blue line) or only one randomly selected sister cell from each pair (red line).

E Difference in variance explained between red and blue traces in panel (D). The black line represents the actual difference; the red line shows a smoothing (moving average = 1.25 h) of the black trace to highlight the overall trend.

To estimate the variance explained by asymmetric division to daughter cells, we repeated the continuous $R^2$ analysis on two sets of the mother–daughter traces used in the previous analysis. The first set contained mother–daughter traces for all pairs of sister cells. In this set, the same mother trace was paired with exactly two daughter cell traces. The second set contained one randomly chosen mother–daughter trace from each sister pair in the first set and thus contained exactly half the number of traces as the first set. We then repeated the analysis for 100 randomly sampled subsets of these two sets of traces to estimate the change in variance explained by adding measurements of OCT4 for sister cells (Fig 4D and E). As expected, we found that including both sister pairs (blue line) weakened the coefficient of determination before cell division, since the same mother cell trace was paired with two different daughter cell traces. After cell division, however, this trend reversed; including both daughter cell traces improved $R^2$ by ~ 5% with a sustained influence of ~ 3%. Although not a major source of variation, this value is nonetheless consistent with the small percentage of sister cells giving rise to different pro-fates (Fig 2A). Taken together, these results indicate that ~ 15% of the final OCT4 variance is accounted for by maternal OCT4 levels. During the inherited period, most of the OCT4 that is predictive of cell fate is conferred through transmission of OCT4 to both daughters (~ 40% of variance explained). This is consistent with the observation that sister cells choose similar fates. A smaller portion (~ 5%) of explanatory power during the inherited period is introduced through asymmetric division. Thus, the combination of maternal and inherited OCT4 expression patterns—which transpire before daughter cells are 1 h old—explain most (~ 60%) of the differences in cells' long-term OCT4 expression levels.

## Discussion

In this study, we developed an endogenous fluorescent reporter for the canonical pluripotency factor OCT4. This reagent allowed non-invasive monitoring of the pluripotent state of human embryonic

first hour after cell division, however, the $R^2$ between current and final OCT4 expression increased sharply to ~ 0.6, suggesting that newly established OCT4 levels in daughter cells had a strong effect on final OCT4 levels. During the remainder of the daughter cell lifetime—the autonomous period—the coefficient of determination rose gradually to identity ($R^2 = 1$) since we defined the last time point in the daughter cell's lifetime to be the final OCT4 expression level.

stem cells and the ability to capture differentiation decisions in real time. By tracking the OCT4 dynamics for individual cells over multiple cell cycle generations, we found that a single cell's decision to differentiate to embryonic mesoderm is largely determined before the differentiation stimulus is presented to cells. Before BMP4 treatment, a cell's preexisting OCT4 levels, pulsing frequency, and cell cycle duration accurately (but not perfectly) predicted the eventual fate decisions of its offspring cells. These results harmonize with studies of mouse ESCs in which cell-to-cell differences in OCT4 expression (Niwa *et al*, 2000; Zeineddine *et al*, 2006; Radzisheuskaya *et al*, 2013; Goolam *et al*, 2016) and degradation kinetics (Plachta *et al*, 2011; Filipczyk *et al*, 2015) are associated with different developmental fate decisions.

One implication of this study is that stem cells possess some form of molecular "memory" that persists over multiple cell cycle generations to influence fate decisions. Interestingly, however, the half-life of OCT4 was found to be ~ 7 h, which is roughly half of an average cell cycle duration. In agreement with this measurement, autocorrelation analysis of single-cell OCT4 traces showed a mixing time (Sigal *et al*, 2006) of ~ 40% of a cell cycle duration (Appendix Fig S11). Thus, the turnover rate of OCT4 protein is faster than the duration of the predictive cell fate memory. This suggests that OCT4 cooperates with other molecular factors to maintain cell identity over the course of multiple cell cycle generations. Indeed, OCT4 is known to influence chromatin conformation (Dixon *et al*, 2015), epigenetic modifications (Singer *et al*, 2014; Bintu *et al*, 2016), and additional pluripotency factors (Goolam *et al*, 2016) through positive feedback mechanism (Boyer *et al*, 2005). Through these mutually reinforcing interactions, it is possible that continual turnover of OCT4 on a short timescale could provide persistent memory over longer time scales (Cheng *et al*, 2008). Since it is well established that even modest differences in OCT4 expression can affect the differentiation potential of stem cells (Nichols *et al*, 1998; Niwa *et al*, 2000; Zeineddine *et al*, 2006; Radzisheuskaya *et al*, 2013), OCT4 is probably both a cause and a consequence of the pluripotent state and, under the experimental conditions studied here, is a significant contributor to the molecular memory system that guides the fate of human stem cells.

Our work also suggests that molecular inheritance of OCT4 is established shortly after cell division. The major portion of OCT4 inheritance (~ 40% of variance explained) is conferred to both daughter cells, whereas a smaller portion (~ 3–5%) is due to differences between sister cells (Fig 4C–E). This finding is consistent with the observation that sister cells typically choose the same fate (Fig 2A). Both of these mechanisms are consistent with a model of mitotic bookmarking in which pluripotency factors such as OCT4 bind tightly to chromatin during mitosis to retain pluripotency gene expression program in daughter cells (Egli *et al*, 2008; Liu *et al*, 2017). It is possible that strong retention of OCT4 through mitosis essentially seals the fate of daughter cells. If true, this would imply that a cell's autonomous decision-making ability may be precluded by inherited factors that bind tightly to chromatin. Our work provides a novel way to analyze the allocation and influence of bookmarking factors over time throughout the process of cell division.

Finally, this study offers a new perspective for the field of developmental systems biology. Our results imply that, under the experimental conditions studied, the fate of an individual stem cell is almost entirely determined by the time that cell is born through inheritance of a pluripotency factor. This idea challenges the view that every cell is a "clean slate" whose fate can be altered by external stimuli. On the contrary, our work shows that the levels of maternal OCT4, even up to two or three generations prior to a differentiation cue, have the ability to prime a stem cell for a particular response to external stimuli. Future work will be necessary to determine the mechanisms that generate and regulate OCT4 variability across a cell population; whether the predictive nature of OCT4 is related to its role in mitotic bookmarking; and whether these mechanisms can be exploited for therapeutic uses of human stem cells.

# Materials and Methods

### Culture and treatment of hESCs

WA09 (H9) hES cell line was purchased from WiCell (Wisconsin) and maintained in mTeSR1 (05850, StemCell Technologies) on growth factor reduced Matrigel (354230, BD). Cells were passaged every 3 days using 0.5% EDTA in PBS.

### Guide RNA and CRISPR/Cas9 cutting vector

The gRNA sequence GTGAAATGAGGGCTTGCGA, targeting the start codon of human *POU5F1* (OCT4), was cloned into pX330 (AddGene) using the standard cloning protocol described by Ran *et al* (2013). The cutting efficiency of the Cas9/OCT4-gRNA was validated with Guide-it Mutation Detection Kit (Takara Bio).

### Donor cassette construction

The 5′ homology arm of OCT4 was amplified out of H9 genomic DNA with the following primers (Fwd: 5′-AAGGTTGGGAAACT GAGGCC-3′, Rev: 5′-GGGAAGGAAGGCGCCCCAAG-3) yielding a 1,114 bp homology arm that was then cloned into the pGEMTEZ plasmid (Promega) followed by the coding sequence for the mCherry fluorescent protein (minus its stop codon) followed by a short linker sequence (TCC GGA TCC) and the start ATG codon for OCT4. The OCT4 gene constituted the 3′ homology arm and was amplified out of H9 genomic DNA with the following primers (Fwd: 5′-ATGGCGGGACACCTGGCTTC-3′, Rev: 5′-AGCTTTCTACAAGG GGTGCC-3′) yielding a 1,082 bp homology arm.

### Introduction of exogenous DNA into H9 cells

H9 cells were cultured on 10-cm dishes and, when 80% confluent, were dissociated using 0.5 mM EDTA. $10 \times 10^6$ cells were resuspended in 800 µl ice-cold PBS containing 25 µg of the OCT4-mCherry donor vector and 25 µg of the guideRNA/Cas9 vector. Cells were electroporated in 100 µl tips (Neon, ThermoFisher Scientific) using program 19 of the optimization protocol (1,050 V, 30 ms, two pulses) and resuspended in mTeSR1 (STEMCELL Technologies) supplemented with Rock inhibitor (S1049, Selleck Chemicals) at a final concentration of 10 µM. When colonies that expressed mCherry reached approximately 20 mm in size, they were marked and picked into Matrigel coated 24-well plates.

### Endogenous OCT4 levels

Endogenous OCT4 levels in H9 wild-type cells and H9 OCT4-mCherry clone 8-2 were determined by antibody staining using a mouse anti-OCT4 antibody (MABD76, EMD Millipore). Immunostaining was performed using standard protocols. Briefly, cells were fixed for 15 min in 4% paraformaldehyde and permeabilized and blocked for 30 min in 5% goat serum with 0.3% Triton X-100 in TBS. Incubation with primary antibody was performed overnight, and the incubation with the secondary antibody (Molecular Probes) was done at room temperature for 45 min. Nuclei were visualized using NucBlue Fixed Cell Stain ready Probes reagent (R37606, Molecular Probes).

### Live-cell imaging

Asynchronous H9 OCT4-mCherry cells were plated on 12-well glass bottom plates (Cellvis) in phenol-red free or clear DMEM/F-12 (Gibco) supplemented with mTeSR1 supplement (05850, STEMCELL Technologies) approximately 24 h before being imaged. Cells were imaged using a Nikon Ti Eclipse microscope operated by NIS Elements software V4.30.02 with an Andor ZYLA 4.2 sCMOS camera and a custom stage enclosure (Okolabs) to ensure constant temperature, humidity, and $CO_2$ levels. Fresh media with or without BMP4 were added every 24 h. Images were flat-field-corrected using NIS Elements.

### Image analysis

A custom ImageJ plugin (available upon request) was used to perform automated segmentation and manually tracking of hESCs. Fluorescence intensity was quantified using an adapted threshold followed by watershed segmentation of the OCT4-mCherry channel. The program tracked the cell ID, parent ID, frame number, and mean intensity and exported this information to MATLAB for analysis.

### Quantitative analysis

All computational methods including lineage analysis and logistic regression are included as Dataset EV1, which includes processed image data and documented MATLAB code used to generate each of the figures.

### OCT4 pulses

OCT4 pulses were identified by finding peaks within single-cell traces of OCT4 expression in individual cells. The code used to identify peaks is found in the getcellpeaks.m in Dataset EV1. Briefly, peaks were required to have a minimum width of 15 min (three frames) and a minimum prominence (i.e., how much the peak stands out due to its intrinsic height and its location relative to other peaks) of 200 a.f.u., which is approximately equal to 1 standard deviation in OCT4 expression levels across individual cells. The relationship between OCT4 pulses and stem cell pro-fate, as reported in the paper, was not sensitive to small variations in these parameter choices.

### OCT4 half-life calculations

We used cyclohexamide treatment to estimate the half-life of OCT4-mCherry to be 7.34 h (Appendix Fig S3). If 50% of OCT4-mCherry is degraded in 7.34 h, then the percentage of OCT4 left after only 5 min is

$$N(t) = N_0 \left(\frac{1}{2}\right)^{\frac{t}{t_{1/2}}}$$

$$N(t) = 100 \left(\frac{1}{2}\right)^{\frac{0.0833}{7.34}}$$

$$N(t) = 99.22$$

### Logistic regression

Logistic regression was performed using the fitglm function in MATLAB using a binomial model and logit link function. The code used to train the model is found in the trainClassifier.m function in Dataset EV1.

**Expanded View** for this article is available online.

### Acknowledgements

We thank Paul Lerou, Galit Lahav, Effie Apostolou, Bobbie Pelham-Webb, Allon Klein, Jean Cook, Paul Maddox, Bill Marzluff, Scott Bultmann, Peijie Sun, Robert Corty, Greg Keele, Will Valdar, and members of the Purvis Lab for helpful discussions and technical suggestions. We thank Caroline Purvis for inspiring development of the OCT4-mCherry reporter. This work was supported by NIH grant DP2-HD091800-01, the W.M. Keck Foundation, and the Loken Stem Cell Fund.

### Author contributions

SCW, RD, ASB, and JC constructed the OCT4-mCherry reporter cell line. SCW, ASB, and RD performed validation studies. SCW and CDD performed live-cell imaging. CDD, RAH, TMZ, MAR, and KMK conducted image analysis and cell tracking. KMK, CDD, RAH, and JEP performed computational analysis. SCW and ASB carried out lineage differentiation experiments. JEP wrote the manuscript with contributions from all authors.

### Conflict of interest

The authors declare that they have no conflict of interest.

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
