## [Review Process File · Molecular Systems Biology]

Inheritance of OCT4 predetermines fate choice in human embryonic stem cells

Samuel C. Wolff, Katarzyna M. Kedziora, Raluca Dumitru, Cierra D. Dungee, Tarek M. Zikry, Adriana S. Beltran, Rachel A. Haggerty, JrGang Cheng, Margaret A. Redick and Jeremy E. Purvis.

Review timeline:

Submission date:	24 th January 2018
Editorial Decision:	20 th February 2018
Revision received:	31 st May 2018
Editorial Decision:	2 nd July 2018
Revision received:	28 th July 2018
Accepted:	30 th July 2018

Editor: Maria Polychronidou.

Transaction Report:

1st Editorial Decision

20th February 2018

Thank you again for submitting your work to Molecular Systems Biology. We have now heard back from the three referees who agreed to evaluate your study. As you will see below, the reviewers acknowledge that the study seems potentially interesting. They raise however a series of concerns, which we would ask you to address in a major revision.

The reviewers' recommendations are rather clear and therefore I think that there is no need to repeat all the points listed below. Reviewers #2 and #3 think that additional analyses need to be performed in order to better support the main conclusions and they provide constructive suggestions in this regard. Please let me know in case you would like to discuss further any of the issues raised by the reviewers.

REVIEWER REPORTS.

Reviewer #1:

The manuscript entitled "Inheritance of OCT4 predetermines fate choice in human embryonic stem cells" reports the study of gene expression inheritance in stem cells by focusing on OCT4, a well known transcription factor that is involved the control of self-renewal. By tagging endogenous OCT4 with mCherry, by means of CRISPR/CAS9 gene editing, the authors could track OCT4 gene expression over time across multiple cell divisions in a population of human embryonic stem cells (hESCs). Using this approach, the authors demonstrated that OCT4 inheritance in hESCs is in most cases predetermined in the mother cells and the different segregation of OCT4 during mitosis is then maintained after cytokinesis. These results provide a novel viewpoint on how heterogeneity and differentiation are established in a stem cell community. In general, the experiments are carefully done and the conclusions are well supported by the data. However, I would recommend a revision of

the manuscript to further improve its strength and clarity.

Major concerns:

1) In the 2D single-cell scatter plot in figure 2F, the authors show the correlation between the frequency of OCT4 bursts and OCT4 expression. However, it is not clear how the burst frequency was calculated.

First: how were the bursts defined? In figure 1G, curves show the presence of multiple bursts of different shape and size (e.g. cell #6 shows one large peak and several small peak).

Second: did the authors define a threshold to distinguish bursts from noise?

Third: How did the authors count the bursts in cell tracks? Did they count them manually or did they use an automatic procedure?

2) In relation to figure 3D, the authors write: "the distribution of OCT4 was NOT perfectly symmetric but instead adopted a bell-shaped distribution that was centered around a mean ratio of 1 ($r/r = 1/1$)". This statement is confusing.

The bell-shaped curve is symmetric by definition and the asymmetry that authors refer to lies in the fact that sometimes cells end up with $r = 5/6$ (or $6/5$). However, the occurrence of these asymmetries is the same for left and right side of the (symmetric) distribution. Please rephrase this section.

3) In the result section the authors write: "OCT4 would be >99% maternal 5 minutes after division" and "that is established in a daughter cell 30 minutes after division is more than 95% maternal".

Where are these percentages derived from? Are they derived from some measurements or from other published papers (in the latter case, add citation)?

Minor concerns:

1) Why are the right-bottom snapshots in figure EV5a black?

2) The legend of figure EV3 is missing.

3) Is the experiment shown in EV9 made in presence or absence of BMP4?

4) Page 9 of the manuscript: correct EV10.

Reviewer #2:

The paper by Wolff et al addresses an important question related to key cell decision to proliferate or differentiate. They investigate the specific question of timing. At what point in time do cells actually "decide"? The key assay used here is a coupling of live cell imaging of an endogenously tagged Oct4 coupled to immunofluorescent staining of marker that can provide information of the final decision. They argue that cells are "predetermined" to go into a specific decision and that this is influenced by asymmetric division of Oct4 during mitosis.

The paper addresses a fundamental question and shows potentially intriguing results. However, I had a few conceptual and technical, that must be addressed before it can be published.

Conceptual points:

1. It is hard for me to understand where the "memory" that they author argue is maintained for a long time is "stored". A simple interpretation of their results would argue that the levels of Oct4 are where the "memory" is stored. But, given the bursty expression and ~8 hours half-life, I can't see how cells can "remember" their state for more than 12-16 hours. If, on the other hand, "memory" is not simply oct4 levels, then I don't understand why the asymmetric division matters so much. This is a key point that has to be resolved. Unless I am missing something, the two claims cannot be both true. One experiment that could help it to analyze the autocorrelation time of Oct4 similar to work by Sigal et al (PMID: 17122776).

2. Given their proposed key role of asymmetric distribution of Oct4 during division, you would expect that most cells will show a "mixed" decision with one daughter differentiating and one proliferating. However, they only see 5% of the cells showing this "mixed" phenotype. If that is the case, I don't understand why the asymmetric distribution can be invoked to act as a key role that generates variability. There could be some other factor, for example, an epigenetic effect similar to Singer et al (PMID: 25038413) that is where "memory" is. This would make more sense if most sisters are choosing the same fate despite the different distribution of Oct4.

3. The authors imply that cell decision was made long before the BMP stimulus: "were evident as early as 2 days before the differentiation stimulus was presented, were associated with alternate cell fate decisions". However, in when they actually quantify correct decision in figure 4, the correct fate prediction from the entire Oct4 trajectory prior to division, they only get ~80% correct fate. Given the class imbalance and the fact that ~75% of the cells chose the same fate, it is unclear to me is 80% is that different from random. Might be statistically different, but the effect size is very small. This is supported by the fact that most of the "errors" of the logistic model are in one class (differentiation, Figure 4C) and not proliferation. In the end, I am left unsure what is really the effect size of knowing Oct4 levels at different times and therefore it cast doubt on the main claims in this work.

Technical points:

1. Given the key role of Thalf in this work and the fact that the authors have a fluorescently tagged protein, I was surprised that they chose a CHX treatment to measure protein half-life. Blocking ribosomes can do a lot of things to cells... The author should redo that experiment using bleaching similar to what was done in (PMID: 21233346).

2. The claims related to asymmetric distribution should be strengthened by a simple control. They should measure something else that we know is equally distributed (DNA using DAPI, histone using IF) and using the same image analysis mask show histograms similar to Figure 3D.

Minor points:

1. The use of "Burst" is problematic. In the context of gene expression noise, it caused me to think that they might imply a transcriptional burst (i.e. DNA off->on kind of event). However, these events cannot be measured by simply counting peaks in a reporter with a ~ 8-hour half-life (or potentially longer memory, see above). Given that burstiness is not a key part of their thesis, don't think that this is what they mean, but I think it would be better if they don't use that term and talk about measurable pulses or something like that.

2. I would recommend changing data presentation in Figure 4. The heatmaps are hard to see, and in any case, they used an average of time series in their regression analysis. Simply showing logistic regression will be better. Or, come up with a better way to quantify decision that takes into account the class imbalance. They way Figure 4D is shown making one thing that there is a bit effect when it is not clear to me that there is one.

Reviewer #3:

Summary

Human embryonic stem cells showed heterogeneous cell fates (self-renewal vs. differentiation) after 24 hours of BMP4 treatment (differentiation stimulus). To understand whether the cell-fate decision was predetermined before stimulus or autonomous after stimulus, the authors established a live-cell reporter of endogenous OCT4. By monitoring OCT4 levels starting from 42 hours before stimulus to 24 hours after stimulus, the authors found that cell fate (self-renewal vs. differentiation) was largely determined before the differentiation stimulus, and can be predicted by preexisting cellular state, especially OCT4 levels. The authors further showed that the significant differences in OCT levels between sister-cell pairs were established immediately after cell division. This asymmetric distribution of OCT4 between daughter-cell pairs led to sustained differences in OCT4 as well as differences in final fate choice. Together, these results suggested that cell fate was mostly determined by the time of birth.

General remarks

This study investigates the interesting hypothesis that the capacity of stem cells to differentiate might be largely determined before they receive differentiation stimulus. If this hypothesis holds true, the concept of stem-cell pluripotency would be advanced from individual-cell focus to incorporation of lineage history. The authors have developed a nice experimental system for tracking OCT4 expression in stem cells and linking expression to renewal/differentiation fate. However, the analyses provided in this study were not direct or sufficient to support the major conclusions. Inconsistency in method descriptions made the main result (Figure 4) difficult to interpret, and the logic of the story was in key places hard to follow.

Major points

Three key conclusions of this study are assessed separately below.

The first key conclusion is that cell fate (self-renewal vs. differentiation) is largely determined before the differentiation stimulus, and can be predicted by a cell's preexisting OCT4 levels, bursting frequency, and cell cycle duration (Figure 2). The significance of the measured differences between different pro-fates was used to support this conclusion (Figure 2C-E). However, this is not directly related to prediction. A more direct measure would be the prediction accuracy of a classifier (e.g. logistic regression) built using these cell-state properties. Furthermore, only OCT4 levels showed clear difference between the two fates that this study mainly focused on-self-renewal and differentiation. Burst frequency did not show significant differences between these two fates, and cell cycle duration only had small size effect. Therefore, without a direct measure of prediction accuracy, it is not convincing that cell fate can be predicted by a cell's preexisting OCT4 levels, bursting frequency, and cell cycle duration.

The second key conclusion is that the significant differences in OCT4 levels between sister-cell pairs were established at the moment of cell division (Figure 3D) and can predict final OCT4 levels (Figure 3E). Overall, the section of the paper around Figure 3 is confusing. Early text (Figure 3A,B) asserts that OCT4 is similar for sister pairs yet later text (Figure 3D) asserts that there are "significant differences". It was hard to understand the rationale behind these opposing claims. In any case, the authors showed the distributions of OCT4 ratio between daughter cells at different time points and correlation between inherited OCT4 ratio (30 minutes after cytokinesis) and final OCT4 ratio. However, the OCT4 measurements used for the ratio calculation are not consistent in the legends of Figure 3D (mean OCT4) and EV9 (cumulative sum of mean OCT4), which needs to be clarified. In addition, showing asymmetric distribution of OCT4 to daughter cells at the moment of cell division (Figure 3D, 5 min) is not sufficient to support the conclusion that the significant differences in OCT4 levels between sister-cell pairs were established at the moment of cell division. A more convincing analysis would be breaking down the total variation OCT4 levels between sister-cell pairs by different time points (or periods), and show the percentage of variation explained by each time point (or period). Furthermore, correlation between early and final OCT4 ratio does not directly support the conclusion that inherited OCT4 levels can predict final OCT4 levels. A more direct assessment of this conclusion would be using a linear regression (or any other prediction model) to predict final OCT4 levels (not ratio) using early OCT4 levels (not ratio), and report R-squared instead of correlation. Finally, from the distributions shown in Figure 3D, it is not clear whether a sister pair with a ratio >1 at one time point could have a ratio <1 at a later time point; Figure 3E tries to address this, however it appears as though many cell pairs (2nd and 4th quadrants) switch ratios. Thus we are left to wonder how "sustained" (word used in abstract) this OCT4 difference at birth really is.

The third key conclusion is that cell fate is mostly determined by the time of birth (Figure 4). However, the inconsistency in method description made this figure difficult to interpret. According to Figure 4A and its legend, maternal, inherited and autonomous OCT4 were calculated using three separate time periods of a mother-daughter cell trace. However, in the main text ("Autonomous OCT4 levels include both maternal and inherited levels as well as those experienced during the remainder of the cell's own lifetime.") and Figure 4C and 4D, autonomous OCT4 seems to include inherited OCT4, which also includes maternal OCT4. The definition is crucial and needs to be clarified.

The main result to support the third key conclusion was Figure 4D. However, if data used to study inherited OCT4 includes the maternal OCT4 period, the higher prediction accuracy obtained by using inherited OCT4 cannot be exclusively attributed to the OCT4 levels established within 30

minutes of cell division. Unless the logistic regression model showed that most of the coefficients associated with maternal OCT4 were essentially zero, the third key conclusion was not well supported. Alternatively, three logistic regression models can be built using non-overlapping OCT4 information (as defined in Figure 4A). Then the prediction accuracy obtained by these three logistic regression models could be compared to assess the contribution of different time periods. In addition, the data used in this analysis was unbalanced, with 77% (286/370) of cells having self-renewal fate. Thus, the baseline of prediction accuracy would be 77% (naively predicting all cells to have a self-renewal fate would reach 77% accuracy). With this in mind, the 79% accuracy obtained by using only maternal OCT4 levels before cell birth was not surprising, as opposed to what was written in the main text ("By considering only maternal OCT4 levels before cell birth, the logistic regression model was already 79% correct in predicting cellular fate..."). Furthermore, the authors described that the prediction accuracy rose from 79% to 82% (over the lifetime of the mother cell) to 88% (within 10 minutes of daughter cell birth) and finally reached 93% (within 6 hours of birth). Based on this result, they concluded that "...differentiation of human embryonic stem cells is almost entirely predetermined both by maternal and mitotic OCT4 expression dynamics, whereas the contribution of autonomous signaling is relatively minor". However, a close look revealed that maternal and mitotic OCT4 provided 3% and 6% improvement, respectively, and autonomous signaling provided another 5%. It is not clear why 5% is a minor contribution compared to 9% (3%+6%).

Minor points

1. Missing scale bar in Figure 1B.
2. Missing legend for Figure EV3.
3. How was burst quantified in Figure 2D?
4. "...these differences are stabilized within the first 10-20 minutes of cellular lifetime (Figure EV10)." However, Figure EV10 was not provided.
5. Experiment that generated the data set used in Figure 4 is not clear.
6. "We considered 304 cells for which we captured their complete cell-cycle lifetime (from birth to division) before BMP4 treatment and for which the pro-fate of self-renewal (286 cells) or differentiation (84 cells) was definitively known". However, $304 \neq 286 + 84$.
7. Error occurred when running the provided code: "Undefined function or variable 'dscatter'. Error in plotoct4vcdx2 (line 5)".

We thank the referees for making valuable suggestions that have spurred us to provide stronger support for the conclusions of this study and, in addition, to improve its clarity. This document includes a complete transcript of their comments along with our responses.

Reviewer #1:

The manuscript entitled "Inheritance of OCT4 predetermines fate choice in human embryonic stem cells" reports the study of gene expression inheritance in stem cells by focusing on OCT4, a well-known transcription factor that is involved the control of self-renewal. By tagging endogenous OCT4 with mCherry, by means of CRISPR/CAS9 gene editing, the authors could track OCT4 gene expression over time across multiple cell divisions in a population of human embryonic stem cells (hESCs). Using this approach, the authors demonstrated that OCT4 inheritance in hESCs is in most cases predetermined in the mother cells and the different segregation of OCT4 during mitosis is then maintained after cytokinesis. These results provide a novel viewpoint on how heterogeneity and differentiation are established in a stem cell community. In general, the experiments are carefully done and the conclusions are well supported by the data. However, I would recommend a revision of the manuscript to further improve its strength and clarity.

We would like to thank the reviewer for the positive comments and address below the specific points raised.

Major concerns:

1) In the 2D single-cell scatter plot in figure 2F, the authors show the correlation between the frequency of OCT4 bursts and OCT4 expression. However, it is not clear how the burst frequency was calculated. First: how were the bursts defined? In figure 1G, curves show the presence of multiple bursts of different shape and size (e.g. cell #6 shows one large peak and several small peak). Second: did the authors define a threshold to distinguish bursts from noise? Third: How did the authors count the bursts in cell tracks? Did they count them manually or did they use an automatic procedure?

OCT4 bursts (now called "pulses" per Reviewer 2's suggestion; please see below) were calculated by finding peaks within single-cell traces of OCT4 expression in individual cells. The code used to identify peaks is found in the `getcellpeaks.m` in Dataset EV1. Briefly, peaks were required to have a minimum width of 15 min (3 frames) and a minimum prominence (i.e., how much the peak stands out due to its intrinsic height and its location relative to other peaks) of 200 a.f.u., which is approximately equal to 1 standard deviation in OCT4 expression levels across individual cells. The relationship between OCT4 pulses and stem cell pro-fate were not sensitive to small variations in these parameter choices.

We have updated the **Materials and Methods** section to clarify how OCT4 pulses were calculated.

2) In relation to figure 3D, the authors write: "the distribution of OCT4 was NOT perfectly symmetric but instead adopted a bell-shaped distribution that was centered around a mean ratio of 1 ($r = 1/1$)". This statement is confusing. The bell-shaped curve is symmetric by definition and the asymmetry that authors refer to lies in the fact that sometimes cells end up with $r = 5/6$ (or $6/5$). However, the occurrence of these asymmetries is the same for left and right side of the (symmetric) distribution. Please rephrase this section.

This statement was indeed confusing. We have revised the text to read, "Comparison of OCT4-mCherry intensity between daughter cells revealed that OCT4 was not equally allocated during cell division. Instead, the ratio of OCT4 between daughters showed a bell-shaped distribution with a central tendency toward a ratio of 1 ($r = 1/1$).". In addition, we have modified the figure legend to read, "Ratios for both sister cells (r and $1/r$) are plotted to emphasize central tendency toward $r = 1/1$ "

3) In the results section the authors write: "OCT4 would be >99% maternal 5 minutes after division" and "that is established in a daughter cell 30 minutes after division is more than 95% maternal". Where are these percentages derived from? Are they derived from some measurements or from other published papers (in the latter case, add citation)?

The claim that OCT4 is >99% maternal 5 minutes after division is based on a calculation of the half-life of OCT4 protein levels. In Appendix Fig S3 (previously Figure EV3), we used cyclohexamide treatment to estimate the half-life of OCT4-mCherry to be 7.24 h. If 50% of OCT4-mCherry is degraded in 7.24 h, then the percentage of OCT4 left after only 5 minutes is:

$$N(t) = N_0 \left(\frac{1}{2} \right)^{\frac{t}{t_{1/2}}}$$
$$N(t) = 100 \left(\frac{1}{2} \right)^{\frac{0.0833}{7.24}}$$
$$N(t) = 99.21$$

Similarly, the percentage of maternal OCT4 remaining after 30 min is 95.3%. We should have made this calculation more explicit in the text. We have revised the text to read, "...because the half-life of OCT4 was calculated to be 7.24 h (**Appendix Figure S3**), the OCT4 protein present in a daughter cell 5 minutes after division is estimated to be >99% maternal. Thus, it is unlikely that asymmetric ratios were due to stochastic differences in protein degradation during the first 5 minutes of daughter cell lifetime."

We have updated the **Materials and Methods** section to clarify how OCT4 half-life calculations were performed.

Minor concerns:

1) Why are the right-bottom snapshots in figure EV5a black?

The two bottom panels in the fourth column of Figure EV5a (now **Appendix Figure S5A**) are images of regions of the well that do not contain any cells. These two replicate images serve as negative controls indicating how much light would be expected if the entire fluorescent signal in the cells was lost due to photobleaching.

We have corrected the legend for **Appendix Figure S5** to properly describe these two panels.

2) The legend of figure EV3 is missing.

We apologize for this mistake. Please find a complete legend to **Appendix Figure S3** in the revised manuscript **Appendix**.

3) Is the experiment shown in EV9 made in presence or absence of BMP4?

The original Figure EV9 (now **Appendix Figure S9**) included cells both before and after BMP4 treatment. To be more clear and consistent with Figure 3, we have corrected **Appendix Figure S9** to include a separate analysis of cells with full cell-cycle lifetimes before (Fig S9A-B) and after (Fig S9C-D) treatment with BMP4. We have also included more details in the figure legend to make the analysis steps clear. Finally, we have revised the MATLAB code used to generate Fig S9 (`plotsharecorr.m`) with additional comments and labels.

4) Page 9 of the manuscript: correct EV10.

This error has been corrected.

Reviewer #2:

The paper by Wolff et al. addresses an important question related to key cell decision to proliferate or differentiate. They investigate the specific question of timing. At what point in time do cells actually "decide"? The key assay used here is a coupling of live cell imaging of an endogenously tagged Oct4 coupled to immunofluorescent staining of marker that can provide information of the final decision. They argue that cells are "predetermined" to go into a specific decision and that this is influenced by asymmetric division of Oct4 during mitosis.

The paper addresses a fundamental question and shows potentially intriguing results. However, I had a few conceptual and technical concerns that must be addressed before it can be published.

We would like to thank the reviewer for the positive comments and address her/his specific comments below.

Conceptual points:

1. It is hard for me to understand where the "memory" that the authors argue is maintained for a long time is "stored". A simple interpretation of their results would argue that the levels of Oct4 are where the "memory" is stored. But, given the bursty expression and ~8 hours half-life, I can't see how cells can "remember" their state for more than 12-16 hours. If, on the other hand, "memory" is not simply oct4 levels, then I don't understand why the asymmetric division matters so much. This is a key point that has to be resolved. Unless I am missing something, the two claims cannot be both true. One experiment that could help is to analyze the autocorrelation time of Oct4 similar to work by Sigal et al (PMID: 17122776).

The reviewer raises an excellent point. While OCT4 levels can certainly predict cell fate (Fig. 2C), is OCT4 truly the original source of memory that defines and predetermines a cell's fate? This is a difficult question to answer. Recent studies have demonstrated that molecular memory can persist over multiple cell cycle generations¹⁻³. However, this memory need not be the same as the original factors and events that provoked the memory. For example, DNA damage occurring in a mother cell is not necessarily detectable in its daughter cells because the lesions have presumably been repaired. However, other molecular factors generated by the damage, such as cyclinD1 and p53, constitute a transient molecular memory that is still able to alter the daughter cell's fate².

A similar situation is likely to exist in our system. The original molecular differences predisposing certain cells to differentiation probably include additional factors besides OCT4 such as chromatin conformation⁴ epigenetic modifications^{3,5} and the expression of additional pluripotency factors⁶. The situation is further complicated by known positive feedback interactions between OCT4 and these factors⁷. Through positive feedback, even continual turnover of OCT4 on a short timescale could provide persistent memory over much longer time scales⁸. One round of OCT4 expression would lead to future rounds of OCT4 expression. In other words, the same molecular copies of OCT4 need not be long-lived in order to provide long-term memory via OCT4.

It is well established that OCT4 plays a causal role in determining pluripotency. Numerous studies have shown that even modest differences in OCT4 expression can affect the differentiation potential of stem cells⁹⁻¹². Thus, OCT4 is probably both a cause and a consequence of the pluripotent state and constitutes some portion of the molecular memory that guides the fate of future stem cell generations.

In the revised manuscript, we address this concern more fully through additional experiments and analyses. First, we performed the autocorrelation analysis suggested by the reviewer.

Single-cell traces over multiple cell cycle generations were stretched to a common axis and autocorrelation across cells was calculated as described in Sigal et al., 2006¹. This figure has been included as **Appendix Figure S11** in the revised manuscript.

Interestingly, the mixing-time of OCT4 was found to be ~40% of a cell cycle, or ~5.8 h. This number is generally consistent with the half-life of OCT4 determined by cyclohexamide treatment (7.24 h). These measurements suggest that OCT4 turns over more rapidly than the memory actually lasts (several cell cycle generations). Thus, OCT4 must be cooperating with other molecular factors to maintain cell identity over several generations. As referenced earlier, it is well established that OCT4 levels have a strong effect on pluripotency. The novel aspect of this study is defining when (i.e., at what point in a cell's history), and to what extent, a cell's OCT4 level becomes determinative.

We further address the reviewer's concern about the importance of asymmetric distribution to daughter cells in light of the measured half-life of OCT4. Following a suggestion from Reviewer #3 (see below), we quantify the variance explained by asymmetric division as a function of the time since cell division (see revised **Figure 4**). Mother-daughter cell pairs were aligned to the time of division and, at each aligned time point, the correlation between the current OCT4 level and the final OCT4 level was calculated. To calculate the variance explained by asymmetric division, we repeated this calculation but removed one of the two daughter cells from each pair (randomly).

We found that asymmetric division accounts for ~3-5% of the variance in final OCT4 levels in daughter cells. Although not negligible, this amount is small compared to the large jump in predictive power observed immediately after division in *both* daughter cells. Taken together, these results indicate that most of the inherited quantity of OCT4 that is predictive of cell fate is conferred to *both* daughters. Thus, the reviewers' critiques have been helpful in refining the claims of this paper in two ways. First, inheritance of OCT4 (the amount established in the first hour of cell birth) has a stronger effect on a cell's final OCT4 level than previously stated. However, the major portion of that inheritance (~40%) is conferred to both daughters (which is consistent with the observation that sister cells choose similar fates); ~15% is accounted for by maternal OCT4 levels; and a relatively minor portion (~4%) is conferred through asymmetric division. We have made revisions to the text throughout the manuscript to reflect this revised claim.

The work by Sigal et al. provides a helpful precedent both for the autocorrelation time as well as the bleach-chase experiment (see below). We have now cited this work in the revised manuscript. We also explore this concern more fully in the revised **Discussion** section.

2. Given their proposed key role of asymmetric distribution of Oct4 during division, you would expect that most cells will show a "mixed" decision with one daughter differentiating and one proliferating. However, they only see 5% of the cells showing this "mixed" phenotype. If that is the case, I don't understand why the asymmetric distribution can be invoked to act as a key role that generates variability. There could be some other factor, for example, an epigenetic effect similar to Singer et al (PMID: 25038413) that is where "memory" is. This would make more sense if most sisters are choosing the same fate despite the different distribution of Oct4.

This concern is addressed in the response to the previous criticism. In short, we have now shown that asymmetric division accounts for ~3-5% of the variability in final OCT4 levels—consistent with the small percentage of sister cells showing different fates.

3. The authors imply that a cell decision was made long before the BMP stimulus: "...were evident as early as 2 days before the differentiation stimulus was presented, were associated with alternate cell fate decisions...". However, when they actually quantify correct decision in figure 4, the correct fate prediction from the entire Oct4 trajectory prior to division, they only get ~80% correct fate. Given the class imbalance and the fact that ~75% of the cells chose the same fate, it is unclear to me that 80% is that different from random. It might be statistically different, but the effect size is very small. This is supported by the fact that most of the "errors" of the logistic model are in one class (differentiation, Figure 4C) and not proliferation. In the end, I am left unsure what is really the effect size of knowing Oct4 levels at different times and therefore it cast doubt on the main claims in this work.

This is a valid concern that was also raised by Reviewer #3 (see below). Since 75% of cells belong to the same class, even a trivial regression model could be correct 75% of the time. Rather than perform logistic regression with this class imbalance, therefore, we performed a more straightforward analysis of the predictive power of OCT4 by performing a correlation between OCT4 levels through cell division (**Figure 4C**). Another key difference in our analysis is that we calculate the predictive power of instantaneous, rather than cumulative, OCT4 levels. This new analysis is impervious to class imbalances and utilizes the full dynamic range of final OCT4 values among daughter cells.

We further address this concern through an additional analysis requested by Reviewer #3 (see below). We performed a logistic regression on the single-cell features of the untreated cells presented in Figure 2 to see whether they could correctly predict pro-fate. Briefly, we identified all untreated cells completing a full cell cycle and trained a classifier to predict the fate decision of their offspring using three predictors: OCT4 level, OCT4 burst frequency, and cell cycle duration. We tested several classifiers, which showed accuracy between 72-86% (**Materials**

and Methods). The results for the logistic regression model (82% accuracy) are shown in the revised Figure 2F and Appendix Figure S7.

Technical points:

1. Given the key role of $t_{1/2}$ in this work and the fact that the authors have a fluorescently tagged protein, I was surprised that they chose a CHX treatment to measure protein half-life. Blocking ribosomes can do a lot of things to cells. The author should redo that experiment using bleaching similar to what was done in (PMID: 21233346).

We thank the reviewer for this suggestion. We conducted bleach-chase experiments on the OCT4-mCherry cell line as described in Sigal et al., 2006. Briefly, 11 fields of view of growing H9 OCT4-mCherry cells were bleached by exposing them to high intensity light for 5 minutes. Bleached and control (12) fields of view were subsequently observed at 10 min intervals for 12 hours. The difference in fluorescence between control and bleached regions was used to calculate the turnover rate of Oct4-mCherry protein. We measured the half-life of 6.78 h which is in agreement with cyclohexamide protein stability assay presented in the manuscript.

(Left) Average dynamics of control (black) and bleached (red) H9 Oct4-mCherry cells. Thin lines represent separate fields of view (>50 cells per field of view). Thick lines show the mean of all fields of view for each condition. (Right) Difference between bleached and control cells plotted on a semilogarithmic scale. The slope corresponds to the protein removal rate $\alpha = -0.102 \text{ h}^{-1}$. The calculated half-life is $t_{1/2} = \ln 2 / \alpha = 6.78$.

2. The claims related to asymmetric distribution should be strengthened by a simple control. They should measure something else that we know is equally distributed (DNA using DAPI, histone using IF) and using the same image analysis mask show histograms similar to Figure 3D.

This is a logical and straightforward suggestion. We now provide new data in which OCT4-mCherry was imaged for 24 h under undifferentiated conditions, after which cells were fixed and stained with DAPI. Beginning with the final frame of the movie, we identified daughter cell pairs that had emerged from a common mother cell no more than 1 hour earlier. We then quantified

the differences in total DAPI and total OCT4 signal between daughter cell pairs captured during the first, second, or third hour after division. These results confirm that the differences in OCT4 are significantly greater than the differences in DAPI signal between sister cell pairs.

Proliferating hESCs were imaged for 24 h, fixed, and stained for DNA content with DAPI. The probability density of DAPI ratio (*blue*) and OCT4-mCherry ratio (*red*) between sister cells that divided ≤ 1 h prior to fixation was calculated for 151 cell pairs using a normal kernel function. A two-sample Kolmogorov-Smirnov test was used to determine significance.

This result is now included in the revised manuscript as **Figure 3E**.

Minor points:

1. The use of "burst" is problematic. In the context of gene expression noise, it caused me to think that they might imply a transcriptional burst (i.e. DNA off->on kind of event). However, these events cannot be measured by simply counting peaks in a reporter with a ~ 8 -hour half-life (or potentially longer memory, see above). Given that burstiness is not a key part of their thesis, don't think that this is what they mean, but I think it would be better if they don't use that term and talk about measurable pulses or something like that.

We agree that the use of the term "burst" could generate some confusion since it is typically used for gene expression. We have changed the term to "pulses" throughout the revised manuscript.

2. I would recommend changing data presentation in Figure 4. The heatmaps are hard to see, and in any case, they used an average of time series in their regression analysis. Simply showing logistic regression will be better. Or, come up with a better way to quantify decision that takes into account the class imbalance. The way Figure 4D is shown makes one think that there is a big effect when it is not clear to me that there is one.

As stated above, we now present the predictive value of OCT4 through a cell's history as the variance explained by the current OCT4 value on the final OCT4 level.

Reviewer #3:

Summary

Human embryonic stem cells showed heterogeneous cell fates (self-renewal vs. differentiation) after 24 hours of BMP4 treatment (differentiation stimulus). To understand whether the cell-fate decision was predetermined before stimulus or autonomous after stimulus, the authors established a live-cell reporter of endogenous OCT4. By monitoring OCT4 levels starting from 42 hours before stimulus to 24 hours after stimulus, the authors found that cell fate (self-renewal vs. differentiation) was largely determined before the differentiation stimulus, and can be predicted by preexisting cellular state, especially OCT4 levels. The authors further showed that the significant differences in OCT levels between sister-cell pairs were established immediately after cell division. This asymmetric distribution of OCT4 between daughter-cell pairs led to sustained differences in OCT4 as well as differences in final fate choice. Together, these results suggested that cell fate was mostly determined by the time of birth.

General remarks

This study investigates the interesting hypothesis that the capacity of stem cells to differentiate might be largely determined before they receive differentiation stimulus. If this hypothesis holds true, the concept of stem-cell pluripotency would be advanced from individual-cell focus to incorporation of lineage history. The authors have developed a nice experimental system for tracking OCT4 expression in stem cells and linking expression to renewal/differentiation fate. However, the analyses provided in this study were not direct or sufficient to support the major conclusions. Inconsistency in method descriptions made the main result (Figure 4) difficult to interpret, and the logic of the story was in key places hard to follow.

We thank the reviewer not only for her/his positive comments but for requiring better evidence to support for the study's main conclusions. We have addressed the specific concerns below.

Major points

Three key conclusions of this study are assessed separately below:

The first key conclusion is that cell fate (self-renewal vs. differentiation) is largely determined before the differentiation stimulus and can be predicted by a cell's preexisting OCT4 levels, bursting frequency, and cell cycle duration (Figure 2). The significance of the measured differences between different pro-fates was used to support this conclusion (Figure 2C-E). However, this is not directly related to prediction. A more direct measure would be the prediction accuracy of a classifier (e.g. logistic regression) built using these cell-state properties. Furthermore, only OCT4 levels showed a clear difference between the two fates that this study mainly focused on-self-renewal and differentiation. Burst frequency did not show significant differences between these two fates, and cell cycle duration only had small size effect. Therefore, without a direct measure of prediction accuracy, it is not convincing that cell fate can be predicted by a cell's preexisting OCT4 levels, bursting frequency, and cell cycle duration.

This is a logical suggestion. In the revised manuscript, we performed logistic regression to determine how well the three single-cell features—OCT4 level, pulse frequency, and cell cycle lifetime—can classify cells as self-renewing or differentiated. The model trained on all 3 features showed 82% accuracy (5-fold cross validating to prevent overfitting and to address class imbalance). Other classifiers had similar accuracies that ranged from 72-86%. In all models, OCT4 level was the strongest predictor of cell fate.

These results are reported in the revised **Figure 2F**.

The second key conclusion is that the significant differences in OCT4 levels between sister-cell pairs were established at the moment of cell division (Figure 3D) and can predict final OCT4 levels (Figure 3E). Overall, the section of the paper around Figure 3 is confusing. Early text (Figure 3A,B) asserts that OCT4 is similar for sister pairs yet later text (Figure 3D) asserts that there are "significant differences". It was hard to understand the rationale behind these opposing claims. In any case, the authors showed the distributions of OCT4 ratio between daughter cells at different time points and correlation between inherited OCT4 ratio (30 minutes after cytokinesis) and final OCT4 ratio. However, the OCT4 measurements used for the ratio calculation are not consistent in the legends of Figure 3D (mean OCT4) and EV9 (cumulative sum of mean OCT4), which needs to be clarified. In addition, showing asymmetric distribution of OCT4 to daughter cells at the moment of cell division (Figure 3D, 5 min) is not sufficient to support the conclusion that the significant differences in OCT4 levels between sister-cell pairs were established at the moment of cell division. A more convincing analysis would be breaking down the total variation OCT4 levels between sister-cell pairs by different time points (or periods), and show the percentage of variation explained by each time point (or period). Furthermore, correlation between early and final OCT4 ratio does not directly support the conclusion that inherited OCT4 levels can predict final OCT4 levels. A more direct assessment of this conclusion would be using a linear regression (or any other prediction model) to predict final OCT4 levels (not ratio) using early OCT4 levels (not ratio), and report R-squared instead of correlation.

A similar concern was raised by Reviewer #2 (please see above) and we repeat our response here. In the revised manuscript, we quantified the variance explained by asymmetric division as a function of the time since cell division (see **Figure 4**). Mother-daughter cell pairs were aligned to the time of division and, at each aligned time point, the correlation between the current OCT4 level and the final OCT4 level (8 h after division) was calculated. To estimate the variance explained by asymmetric division, we repeated this calculation but removed one of the two daughter cells from each pair (randomly).

We found that asymmetric division accounts for ~3-5% of the variance in final OCT4 levels in daughter cells. Although not negligible, this amount is small compared to the large jump in predictive power observed immediately after division in *both* daughter cells. Taken together, these results indicate that most of the inherited quantity of OCT4 that is predictive of cell fate is conferred to both daughters. Thus, the reviewers' critiques have been helpful in refining the claims of this paper in two ways. First, inheritance of OCT4 (the amount established in the first

hour of cell birth) has a stronger effect on a cell's final OCT4 level than previously stated (the total variance explained within first hour is ~60%). However, the major portion of that inheritance (~40%) is conferred to both daughters (which is consistent with the observation that sister cells choose similar fates); ~15% is accounted for by maternal OCT4 levels; and a minor portion (~3-5%) is conferred through asymmetric division. We have made revisions to the text throughout the manuscript to reflect this revised claim.

Finally, from the distributions shown in Figure 3D, it is not clear whether a sister pair with a ratio >1 at one time point could have a ratio <1 at a later time point; Figure 3E tries to address this, however it appears as though many cell pairs (2nd and 4th quadrants) switch ratios. Thus we are left to wonder how "sustained" (word used in abstract) this OCT4 difference at birth really is.

The reviewer makes a good point that the initial ratio of OCT4 is not necessarily sustained for very long. In the analysis in **Figure 4C**, we quantify the variance explained by asymmetric distribution over time and find that it provides some predictive power but does not comprise the majority of predictive power of OCT4 in early cell lifetime.

We have revised the abstract and manuscript by placing less emphasis on the role of asymmetric distribution of OCT4 and more emphasis OCT4 levels established immediately after division.

The third key conclusion is that cell fate is mostly determined by the time of birth (Figure 4). However, the inconsistency in method description made this figure difficult to interpret. According to Figure 4A and its legend, maternal, inherited and autonomous OCT4 were calculated using three separate time periods of a mother-daughter cell trace. However, in the main text ("Autonomous OCT4 levels include both maternal and inherited levels as well as those experienced during the remainder of the cell's own lifetime.") and Figure 4C and 4D, autonomous OCT4 seems to include inherited OCT4, which also includes maternal OCT4. The definition is crucial and needs to be clarified.

We agree that this definition is crucial and that we had confused the terms by making "inherited" levels inclusive of "maternal" levels, and so forth. To simplify the analysis, we now break down the variance explained into three distinct, non-overlapping time periods: maternal (before division), inherited (first hour after division), and autonomous (beyond first hour after division). A complete timeline of variance explained during these three periods is shown in **Figure 4C**. These definitions have also been stated more clearly and consistently in the revised manuscript.

The main result to support the third key conclusion was Figure 4D. However, if data used to study inherited OCT4 includes the maternal OCT4 period, the higher prediction accuracy obtained by using inherited OCT4 cannot be exclusively attributed to the OCT4 levels established within 30 minutes of cell division. Unless the logistic regression model showed that most of the coefficients associated with maternal OCT4 were essentially zero, the third key conclusion was not well supported. Alternatively, three logistic regression models can be built using non-overlapping OCT4 information (as defined in Figure 4A). Then the prediction

accuracy obtained by these three logistic regression models could be compared to assess the contribution of different time periods. In addition, the data used in this analysis was unbalanced, with 77% (286/370) of cells having self-renewal fate. Thus, the baseline of prediction accuracy would be 77% (naively predicting all cells to have a self-renewal fate would reach 77% accuracy). With this in mind, the 79% accuracy obtained by using only maternal OCT4 levels before cell birth was not surprising, as opposed to what was written in the main text ("By considering only maternal OCT4 levels before cell birth, the logistic regression model was already 79% correct in predicting cellular fate..."). Furthermore, the authors described that the prediction accuracy rose from 79% to 82% (over the lifetime of the mother cell) to 88% (within 10 minutes of daughter cell birth) and finally reached 93% (within 6 hours of birth). Based on this result, they concluded that "...differentiation of human embryonic stem cells is almost entirely predetermined both by maternal and mitotic OCT4 expression dynamics, whereas the contribution of autonomous signaling is relatively minor". However, a close look revealed that maternal and mitotic OCT4 provided 3% and 6% improvement, respectively, and autonomous signaling provided another 5%. It is not clear why 5% is a minor contribution compared to 9% (3%+6%).

This is a valid concern that was also raised by Reviewer #2 (see above). Since 75% of cells belong to the same class, even a trivial regression model could be correct 75% of the time. Rather than perform logistic regression with this class imbalance, therefore, we performed a more straightforward analysis of the predictive power of OCT4 by performing a correlation between OCT4 levels through cell division (**Figure 4C**). Another key difference in our analysis is that we calculate the predictive power of instantaneous, rather than cumulative, OCT4 levels. This new analysis is impervious to class imbalances and utilizes the full dynamic range of final OCT4 values among daughter cells.

Minor points

1. Missing scale bar in Figure 1B.

We have added scale bars to **Figure 1B** as well as the other figure panels with digital images of cells (**Figure 1E** and **3C**).

2. Missing legend for Figure EV3.

We have included the legend for Figure EV3 (now **Appendix Figure S3**).

3. How was burst quantified in Figure 2D?

OCT4 bursts (now called "pulses" per Reviewer 2's suggestion; please see above) were calculated by finding peaks within single-cell traces of OCT4 expression in individual cells. The code used to identify peaks is found in the `getcellpeaks.m` in Dataset EV1. Briefly, peaks were required to have a minimum width of 15 min (3 frames) and a minimum prominence (i.e., how much the peak stands out due to its intrinsic height and its location relative to other peaks)

of 200 a.f.u., which is approximately equal to 1 standard deviation in OCT4 expression levels across individual cells. The relationship between OCT4 pulses and stem cell pro-fate, as reported in the paper, were not sensitive to small variations in these parameter choices.

We have updated the **Materials and Methods** section to clarify how OCT4 pulses were quantified.

4. "...these differences are stabilized within the first 10-20 minutes of cellular lifetime (Figure EV10)." However, Figure EV10 was not provided.

The callout to Figure EV10 was intended to call out Figure EV9 (now **Appendix Figure S9**). As per Reviewer 1's comments (see above), Figure S9 now includes a separate analysis of cells with full cell-cycle lifetimes both before (Fig S9A-B) and after (Fig S9C-D) treatment with BMP4. We have also included more details in the figure legend to make the analysis steps clear. Finally, we have revised the MATLAB code used to generate Fig S9 (`plotsharecorr.m`) with additional comments and labels.

5. Experiment that generated the data set used in Figure 4 is not clear.

We have included extended experimental details in the main text and legend to the revised **Figure 4**.

6. "We considered 304 cells for which we captured their complete cell-cycle lifetime (from birth to division) before BMP4 treatment and for which the pro-fate of self-renewal (286 cells) or differentiation (84 cells) was definitively known". However, $304 \neq 286 + 84$.

We have updated and corrected the cell numbers in the revised manuscript.

7. Error occurred when running the provided code: "Undefined function or variable 'dscatter'. Error in plotoct4vcdx2 (line 5)".

We thank the reviewer for testing out the code. The `dscatter.m` function is an open source file from the MATLAB file exchange (<https://www.mathworks.com/matlabcentral/fileexchange/8430-flow-cytometry-data-reader-and-visualization>). We failed to include it as a dependency in the original submission. We have now included `dscatter.m` and tested running the code in multiple environments.

References

- 1 Sigal, A. *et al.* Variability and memory of protein levels in human cells. *Nature* **444**, 643-646 (2006).
- 2 Yang, H. W., Chung, M., Kudo, T. & Meyer, T. Competing memories of mitogen and p53 signalling control cell-cycle entry. *Nature* **549**, 404-408 (2017).

- 3 Bintu, L. *et al.* Dynamics of epigenetic regulation at the single-cell level. *Science* **351**, 720-724 (2016).
- 4 Dixon, J. R. *et al.* Chromatin architecture reorganization during stem cell differentiation. *Nature* **518**, 331-336 (2015).
- 5 Singer, Z. S. *et al.* Dynamic heterogeneity and DNA methylation in embryonic stem cells. *Molecular cell* **55**, 319-331 (2014).
- 6 Goolam, M. *et al.* Heterogeneity in Oct4 and Sox2 Targets Biases Cell Fate in 4-Cell Mouse Embryos. *Cell* **165**, 61-74 (2016).
- 7 Boyer, L. A. *et al.* Core transcriptional regulatory circuitry in human embryonic stem cells. *Cell* **122**, 947-956 (2005).
- 8 Cheng, Z., Liu, F., Zhang, X. P. & Wang, W. Robustness analysis of cellular memory in an autoactivating positive feedback system. *FEBS Lett* **582**, 3776-3782 (2008).
- 9 Nichols, J. *et al.* Formation of pluripotent stem cells in the mammalian embryo depends on the POU transcription factor Oct4. *Cell* **95**, 379-391 (1998).
- 10 Niwa, H., Miyazaki, J. & Smith, A. G. Quantitative expression of Oct-3/4 defines differentiation, dedifferentiation or self-renewal of ES cells. *Nat Genet* **24**, 372-376 (2000).
- 11 Zeineddine, D. *et al.* Oct-3/4 dose dependently regulates specification of embryonic stem cells toward a cardiac lineage and early heart development. *Developmental cell* **11**, 535-546 (2006).
- 12 Radzisheuskaya, A. *et al.* A defined Oct4 level governs cell state transitions of pluripotency entry and differentiation into all embryonic lineages. *Nat Cell Biol* **15**, 579-590 (2013).

Thank you for sending us your revised manuscript. We have now heard back from the two referees who were asked to evaluate your study. The reviewers think that the study has significantly improved as a result of the performed revisions. However, reviewer #3 still raises some concerns regarding the conclusion that a differentiation decision is mostly determined within 1 hour of cell birth during which the levels of Oct4 are established. As such, we would ask you to include further data to further support this conclusion. Alternatively, if such data is not readily available, we would ask you to adjust the related conclusions accordingly to make sure to avoid overstatements that are not directly supported experimentally. Reviewer #3 also raises a few more remaining issues which we would ask you to address in a revision.

REVIEWER REPORTS.

Reviewer #2:

The authors did a fantastic job addressing reviewer comments. I recommend that the paper is to be published as is in MSB

Reviewer #3:

General remarks

The clarity has been significantly improved in the revised manuscript. The authors also addressed most of the raised points. In the revised manuscript, there are two major conclusions. First, the cellular decision to differentiate is largely predetermined and can be predicted by OCT4 levels before the differentiation stimulus is presented to cells. Second, the differentiation decision is mostly determined within 1 hour of cell birth during which final OCT4 levels are mostly established. The provided data and updated analysis generally support the first conclusion. However, the second conclusion is not well-enough supported to determine its validity.

Major points

First, in figure 4, cells were aligned by the time points of birth. However, it is unclear when cells experienced the differentiation stimulus (presumably, in this aligned coordinate, the stimulus time points could be different for each cell). To be consistent with the first conclusion, only cells that were born an hour before the stimulus was applied should be included in this analysis. Second, the prediction of cell fate using OCT4 levels within 1 hour of cell birth was not shown. Only the correlations between OCT4 levels at each time point to final OCT4 levels at 8 hours after cell birth were provided. It was not justified why 8 hours was chosen; thus, it is unclear how well the OCT4 levels within 1 hour of cell birth can predict cell fate at single-cell resolution.

Minor points

1. In the result of figure 2F, the author wrote: "These single-cell features...-were accurate predictors of stem cell fate." However, based on figure 2F, not all features seem to be informative. For example, cell cycle duration does not seem to be predictive (figure 2F, right). To evaluate the predictive power of each single-cell feature, the coefficients of each feature in the logistic regression should be reported together with statistical tests to assess whether they are significantly different from zero. Also, models built using subsets of features should be compared to the model built with all features to assess the necessity of each feature.
2. In the result of figure S9, the author wrote: "..., the ratio of OCT4 established within 15 minutes of division was strongly correlated with the final OCT4 ratio between sisters." However, the data show that the correlations at 15 minutes (2nd or 3rd dot) are quite low (~0.2) across the three conditions. It needs to be justified why the authors think the correlation is strong.
3. In figure 4, R2 is confused with correlation. For example, the author wrote "..., at each aligned time point, calculated the correlation (R2) between the current OCT4 level and the "final" OCT4 level." R2 is the coefficient of determination not correlation.

We thank Reviewer #3 for her/his additional comments to improve the quality of this study. Please see our responses below for how we have addressed the remaining concerns.

Reviewer #3:

General remarks:

The clarity has been significantly improved in the revised manuscript. The authors also addressed most of the raised points. In the revised manuscript, there are two major conclusions. First, the cellular decision to differentiate is largely predetermined and can be predicted by OCT4 levels before the differentiation stimulus is presented to cells. Second, the differentiation decision is mostly determined within 1 hour of cell birth during which final OCT4 levels are mostly established. The provided data and updated analysis generally support the first conclusion. However, the second conclusion is not well-enough supported to determine its validity.

We thank the reviewer for the positive comments and address below the specific points raised.

Major points:

First, in figure 4, cells were aligned by the time points of birth. However, it is unclear when cells experienced the differentiation stimulus (presumably, in this aligned coordinate, the stimulus time points could be different for each cell). To be consistent with the first conclusion, only cells that were born an hour before the stimulus was applied should be included in this analysis. Second, the prediction of cell fate using OCT4 levels within 1 hour of cell birth was not shown. Only the correlations between OCT4 levels at each time point to final OCT4 levels at 8 hours after cell birth were provided. It was not justified why 8 hours was chosen; thus, it is unclear how well the OCT4 levels within 1 hour of cell birth can predict cell fate at single-cell resolution.

We agree with the reviewer that whereas the first claim (i.e., that differentiation is largely predetermined before stimulus is presented to cells) is supported by the updated figures and analysis, the second claim (i.e., that differentiation was mostly determined within 1 hour of cell birth) is not directly demonstrated since the analysis in Figure 4 showed that OCT4 levels within the first hour after birth predict OCT4 levels 8 hours later but not necessarily fate decision directly. Although it is consistent throughout the study that single-cell OCT4 levels correlate strongly with cell fate (e.g., Figure 2B), it is difficult—as the reviewer has pointed out—to directly demonstrate a correlation between a daughter cell's OCT4 levels within the first hour of birth and its final fate assignment since alignment of cells to the time of birth effectively staggers both the treatment timings and fates of the cells.

Thus, we have adjusted the wording in the abstract from:

“As mother cells underwent division, new OCT4 levels established within daughter cells during the first hour of division were strongly predictive of final OCT4 status and cell fate.

to:

“As mother cells underwent division, newly established OCT4 levels in daughter cells rapidly became more predictive of final OCT4 expression status.”

This change in wording tries to capture the phenomenon observed in Figure 4C. We are open to the reviewer or editor providing further suggestions on how to accurately describe these results.

Similarly, we revised the text in the final paragraph of the Results section from:

“Together, the maternal and inherited OCT4 expression patterns explain most (~60%) of the differences in final OCT4 levels. These results indicate that, in this experimental context, the differentiation decision is mostly determined within 1 h of cell birth.”

to:

“Thus, the combination of maternal and inherited OCT4 expression patterns—which transpire before daughter cells are one hour old—explain most (~60%) of the differences in cells’ long-term OCT4 expression levels.”

Finally, we provide justification in the text for the choice of 8 h as a “final” OCT4 status:

Minor points:

1. In the result of figure 2F, the author wrote: "These single-cell features...were accurate predictors of stem cell fate." However, based on figure 2F, not all features seem to be informative. For example, cell cycle duration does not seem to be predictive (figure 2F, right). To evaluate the predictive power of each single-cell feature, the coefficients of each feature in the logistic regression should be reported together with statistical tests to assess whether they are significantly different from zero. Also, models built using subsets of features should be compared to the model built with all features to assess the necessity of each feature.

We thank the reviewer for these suggestions on how to better describe the logistic regression model. We now report the estimated coefficients of each feature, along with their statistics, in Table S1. As expected, OCT4 levels and burst frequency are predictive features whereas cell cycle duration is not. These results have been added to the main text:

“In all classifier models tested, OCT4 level was the strongest predictor of pro-fate ($P = 3.1 \times 10^{-6}$) followed by burst frequency ($P = 0.0081$); cell cycle duration was not a significant predictor of cell fate ($P = 0.85$; Appendix Table S1). Taken together, these results show that undifferentiated hESCs display heterogeneous OCT4 levels, pulse dynamics, and cell cycle durations. Of these single-cell features, OCT4 levels and

dynamics—which were evident more than one day and as many as two cell cycles before the differentiation cue was given—were accurate predictors of stem cell fate.”

We also state in the caption to Appendix Figure S3 that a comparison of all 7 possible models for combinations of 3 predictors revealed that OCT4 level was the strongest predictor of cell fate whereas burst frequency and cell cycle duration had weak predictive power on their own.

2. In the result of figure S9, the author wrote: "..., the ratio of OCT4 established within 15 minutes of division was strongly correlated with the final OCT4 ratio between sisters." However, the data show that the correlations at 15 minutes (2nd or 3rd dot) are quite low (~0.2) across the three conditions. It needs to be justified why the authors think the correlation is strong.

Indeed, the correlation is better described as “significant” ($P < 0.001$) rather than “strong” ($R^2 \approx 0.2$). We have changed the text accordingly.

3. In figure 4, R^2 is confused with correlation. For example, the author wrote "..., at each aligned time point, calculated the correlation (R^2) between the current OCT4 level and the "final" OCT4 level." R^2 is the coefficient of determination not correlation.

Thank you for pointing out this error in nomenclature. We have corrected the figure and its legend.

MOLECULAR SYSTEMS BIOLOGY

Corresponding Author Name: Jeremy Purvis

Manuscript Number: MSB-17-8140R